# A novel bivalent chromatin associates with rapid induction of camalexin biosynthesis genes in response to a pathogen signal in *Arabidopsis*

Kangmei Zhao[1], Deze Kong[2], Benjamin Jin[1], Christina D Smolke[2,3], Seung Yon Rhee[1]*

[1]Carnegie Institution for Science, Department of Plant Biology, Stanford, United States; [2]Department of Bioengineering, Stanford University, Stanford, United States; [3]Chan Zuckerberg Biohub, San Francisco, United States

**Abstract:** Temporal dynamics of gene expression underpin responses to internal and environmental stimuli. In eukaryotes, regulation of gene induction includes changing chromatin states at target genes and recruiting the transcriptional machinery that includes transcription factors. As one of the most potent defense compounds in *Arabidopsis thaliana*, camalexin can be rapidly induced by bacterial and fungal infections. Though several transcription factors controlling camalexin biosynthesis genes have been characterized, how the rapid activation of genes in this pathway upon a pathogen signal is enabled remains unknown. By combining publicly available epigenomic data with in vivo chromatin modification mapping, we found that camalexin biosynthesis genes are marked with two epigenetic modifications with opposite effects on gene expression, trimethylation of lysine 27 of histone 3 (H3K27me3) (repression) and acetylation of lysine 18 of histone 3 (H3K18ac) (activation), to form a previously uncharacterized type of bivalent chromatin. Mutants with reduced H3K27me3 or H3K18ac suggested that both modifications were required to determine the timing of gene expression and metabolite accumulation at an early stage of the stress response. Our study indicates that the H3K27me3-H3K18ac bivalent chromatin, which we name as kairostat, plays an important role in controlling the timely induction of gene expression upon stress stimuli in plants.

*For correspondence:
srhee@CarnegieScience.edu

Competing interest: The authors declare that no competing interests exist.

## Introduction

Plants produce many metabolites that are essential for responding to environmental cues, the expression of which requires tight temporal control (*Barco and Clay, 2019*; *Grotewold, 2005*; *Martin et al., 2010*). One of the most prominent defense metabolites from *Arabidopsis thaliana* is camalexin. Camalexin is an indole alkaloid derived from tryptophan and can be induced rapidly by various biotic and abiotic stimuli in *Arabidopsis* (*Stotz et al., 2011*). Essential biosynthetic enzymes belong to the cytochrome P450 family, including *CYP79B2/3, CYP71A12/13, and PHYTOALEXIN DEFICIENT 3 (PAD3)* (*Klein et al., 2013*; *Schuhegger et al., 2006*). The expression of these enzymatic genes can be rapidly induced by various abiotic and biotic stresses, and pathogen-derived substances such as flagellin 22 (FLG22) and oligogalacturonides (*Denoux et al., 2008*). Previous studies revealed complex transcriptional and translational control of camalexin biosynthesis genes (*Frerigmann et al., 2016*; *Stahl et al., 2016*; *Zhou et al., 2020*; *Birkenbihl et al., 2012*; *Frerigmann et al., 2015*; *Mao et al., 2011*). At the transcriptional regulation level, transcription factors from the MYB family, including MYB34, MYB51, and MYB122, promote camalexin biosynthesis in response to *Pseudomonas syringae* infection (*Stahl et al., 2016*; *Frerigmann et al., 2015*). WRKY33 functions as an activator and directly binds to the

**eLife digest** In the fight against harmful fungi and bacteria, plants have an arsenal of chemicals at their disposal. For instance, species in the crucifer family – which includes mustard, cabbages and the model plant *Arabidopsis thaliana* – can defend themselves with camalexin, a compound produced soon after the plant receives signals from its attacker. What controls this precise timing, however, is still unclear.

For the genes that rule the production of camalexin to be 'read', interpreted, and ultimately converted into proteins, their DNA sequences first need to be physically accessible to the cell. This availability is controlled, in part, by adding or removing chemical groups onto histones, the spool-like structures which DNA wraps around. These precisely controlled modifications ultimately help to activate or repress a gene. Sometimes, activating and inhibiting chemical groups can be present in the same location, creating what is known as a bivalent chromatin domain.

Zhao et al. investigated whether histone modifications regulate when *A. thaliana* produces camalexin in response to an attack. A combination of bioinformatics and experimental approaches highlighted two chemical modifications (one repressive, the other activating) which were present on the same histone, forming a previously unknown bivalent chromatin domain. Mutant plants which did not carry these modifications could not produce camalexin at the right time. Further experiments showed that under normal conditions, both histone modifications were present. However, when the plant was under attack, the level of repressive and activating modifications respectively decreased and increased, leading to gene activation.

Together, the results by Zhao et al. suggest that both histone modifications are required for camalexin genes to respond appropriately to signals from a harmful agent. A deeper understanding of this new mechanism could, in turn, allow scientists to engineer crops that are better at resisting disease.

promoters of camalexin biosynthesis genes (*Birkenbihl et al., 2012*). Besides transcription factors, CALCIUM-DEPENDENT PROTEIN KINASE (CPK)5/6 and MAPK3/6 can phosphorylate WRKY33 to enhance promoter binding and transactivation (*Zhou et al., 2020*; *Mao et al., 2011*). At the translational level, ribosome footprinting showed that genes involved in camalexin biosynthesis, CYP79B2 and CYP79B3, also increased translational efficiency under pattern triggered immunity (*Xu et al., 2017*; *Yoo et al., 2020*). Despite the rich knowledge of these upstream regulators of the camalexin biosynthetic pathway, it remains unknown how the rapid induction upon a pathogen signal is enabled.

The accessibility of target gene regions to transcription factors is determined by the dynamics of chromatin states in eukaryotic cells (*Stricker et al., 2017*). Chromatin states are controlled by epigenetic modifications that influence nucleosome accessibility (*Vihervaara et al., 2018*). Epigenetic modifications constitute various covalent decoration of chemical groups to histones and DNA, which are associated with promoting or repressing gene expression by altering chromatin accessibility to transcription factors (*Stricker et al., 2017*; *Henikoff and Greally, 2016*; *Venkatesh and Workman, 2015*). For example, trimethylation of lysine 27 of histone 3 (H3K27me3), established by the polycomb repressive complex 2, is associated with repressing gene expression (*Carter et al., 2018*). H3K27me3 represses gene expression by increasing chromatin condensation and limiting the recruitment of transcription factors and other components of the transcriptional machinery (*Aranda et al., 2015*). Trimethylation of lysine 4 of histone 3 (H3K4me3) is marked at actively transcribed genes (*Zhang et al., 2009*), which activates gene expression by promoting the recruitment of transcription initiation factors to promoters of target genes (*Lauberth et al., 2013*).

Epigenetic marks that play opposite roles on gene expression can co-localize at the same gene regions to form bivalent chromatins (*Bernstein et al., 2006*; *Voigt et al., 2013*). Bivalent chromatins were initially observed to be associated with genes involved in cell differentiation in human embryonic stem cells and the most reported variant is formed by H3K27me3 and H3K4me3 (*Bernstein et al., 2006*; *Dattani et al., 2018*; *Sachs et al., 2013*). Bivalent chromatins also exist in other mammalian cell types and can be formed by other epigenetic marks, including additional activating (H3K4me1, H3K4me2, H3K36me3, H3K9ac, H3K27ac, H3K14ac, H4K18ac) and repressive (H3K9me2, H3K9me3, H4K20me3, 5Mc) marks (*van der Velde et al., 2021*; *Xu and Kidder, 2018*; *Charlet et al., 2016*; *Matsumura et al., 2015*; *Roy and Sridharan, 2017*). In plants, the H3K27m3-H3K4me3 bivalent

chromatin has been observed at *FLOWERING LOCUS C* (*FLC*) (*Jiang et al., 2008*) in *A. thaliana* and genes associated with cold stress in potato tuber (*Zeng et al., 2019*). Recently, a novel bivalent chromatin formed by H3K27me3 and H3K4me1 was identified in *Brassica napus* (*Zhang et al., 2021*). Therefore, bivalent chromatins can be formed by different pairs of epigenetic marks.

Despite many reports on the existence of bivalent chromatins, much less is known about their biological roles. The only function that has been proposed for a bivalent chromatin is to poise the expression of developmental genes in stem cells for rapid activation into differentiation (*Bernstein et al., 2006*). This hypothesis has been tested and supported by studies only on mammalian embryonic stem cells or cancer cells (*Deng et al., 2013*; *Lima-Fernandes et al., 2019*; *Jia et al., 2012*). The 'poise genes for transcriptional activation' role of bivalent chromatin has not yet been tested on a whole organism. And, to our knowledge, no other role has been implicated for bivalent chromatins.

Here, by integrating epigenomic profiles with a genome-wide metabolic network in *Arabidopsis*, we found that two epigenetic marks playing opposite roles on gene expression, H3K27me3 and H3K18ac, were enriched at camalexin biosynthesis and several other specialized metabolic pathways. Focusing on the camalexin biosynthesis pathway in this study, sequential chromatin immunoprecipitation (SeqChIP)-PCR confirmed that these two marks were co-localized in vivo and formed a previously uncharacterized type of bivalent chromatin. Mutants defective in H3K27m3 and H3K18ac suggested that both modifications were required to determine the timely induction of camalexin biosynthetic gene expression and camalexin accumulation under FLG22 treatment. This study revealed a new type of bivalent chromatin whose function may be to determine the timely induction of defense-compound biosynthesis in response to a pathogen signal.

## Results

Camalexin biosynthesis pathway is one of the best characterized pathways associated with specialized metabolism in *Arabidopsis* (*Stotz et al., 2011*; *Frerigmann et al., 2016*). We identified epigenetic marks enriched in the camalexin biosynthesis pathway using an omics data-driven approach by integrating 16 publicly available epigenomic profiles with *Arabidopsis* genome-scale metabolic network annotation. High-resolution epigenomic profiles including histone variants, DNA methylation and histone modifications were obtained from seedlings grown under similar conditions (*Wang et al., 2015*; *Luo et al., 2013*; *Roudier et al., 2011*). To map these epigenetic marks on metabolic domains, pathways, and genes, we used genome-wide functional annotations of metabolism (*Schläpfer et al., 2017*). Enrichment analysis at the metabolic domain level revealed distinct patterns, including specialized metabolism being enriched with both a repression mark H3K27me3 and an activation mark H3K18ac (*Figure 1A*). Correlation of H3K27me3 and H3K18ac marks on the *Arabidopsis* genome had been reported previously (*Luo et al., 2013*), which we also observed using total metabolic genes (Pearson's correlation coefficient 0.36, t-test p-value 4.6E-11) (*Figure 1—figure supplement 1A*). To determine if specialized metabolic genes drove the correlation of these two marks on metabolic genes, we removed specialized metabolic genes from the total metabolic genes and reanalyzed the correlation patterns. We observed decreased correlation between H3K27me3 and H3K18ac, indicating that specialized metabolic genes substantially contributed to the correlation of these two marks on metabolic genes (Pearson's correlation coefficient 0.11, t-test p-value 3.5E-02) (*Figure 1—figure supplement 1B*). To determine how prevalent the H3K27me3-H3K18ac association was for specialized metabolic genes, we counted the number of genes marked by both modifications. We found that 37 % (324 out of 887) of specialized metabolic genes were marked by both H3K27me3 and H3K81ac (*Figure 1B*). We then asked in which specialized metabolic pathways these two marks were significantly enriched by selecting the pathways that were annotated to specialized metabolism based on the Plant Metabolic Network database. To increase the statistical power, we only included the specialized metabolic pathways containing at least 10 genes, which resulted in 42 pathways. Among them, we found several pathways, including camalexin and glucosinolate biosynthesis pathways, to be significantly enriched with both epigenetic marks (*Figure 1—figure supplement 2*, *Supplementary file 1*). To decipher the nature and function of the H3K27me3-H3K18ac dual association on controlling gene expression, we focused our investigation on the camalexin biosynthetic genes.

We developed two mutually non-exclusive hypotheses about the role of H3K27me3 and H3K18ac on the expression of camalexin biosynthesis genes: (1) H3K27me3 and H3K18ac are co-localized on the same genes in the same cell and control their gene expression induction; or (2) H3K27me3 and

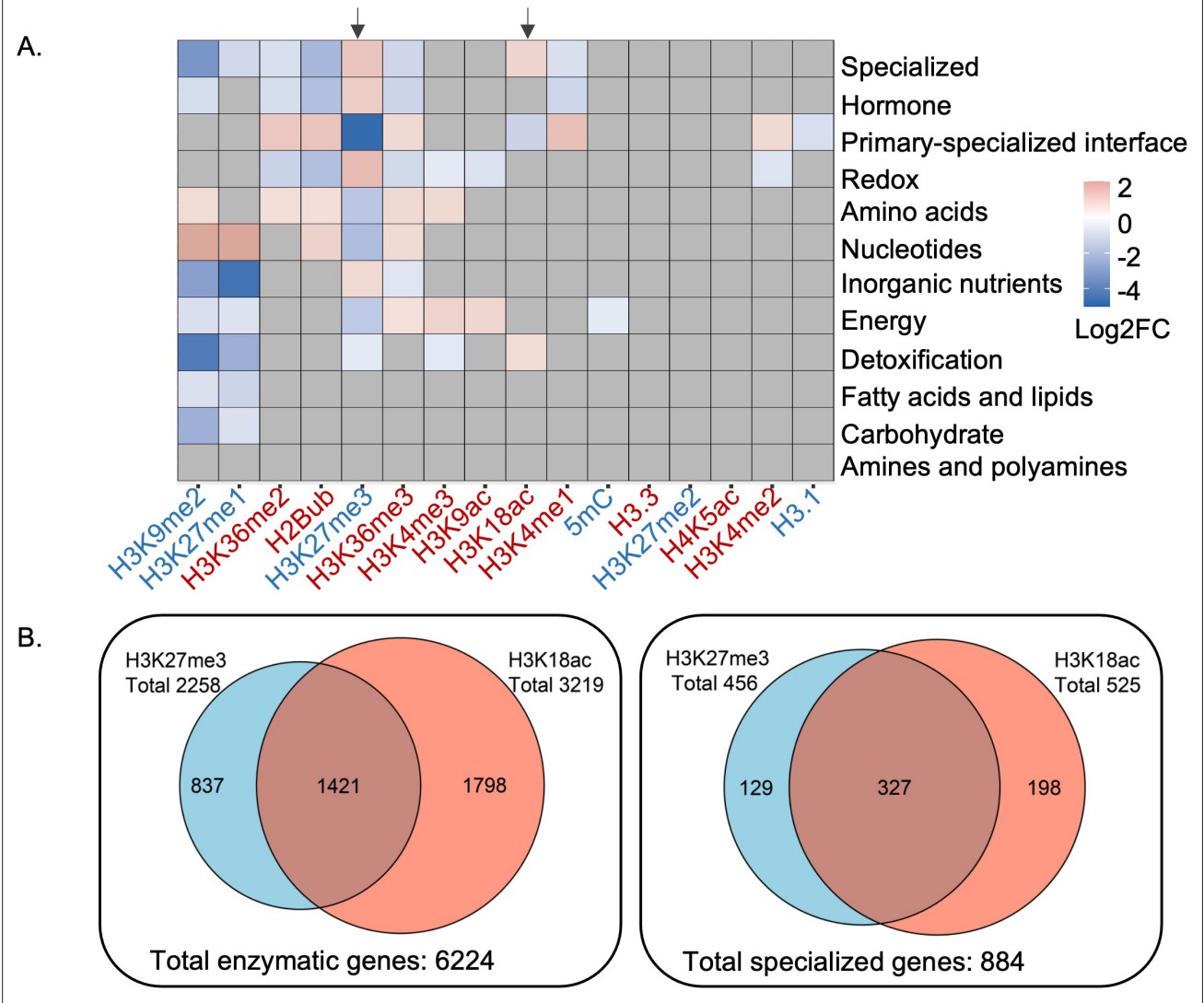

**Figure 1.** Patterns of epigenetic modification across metabolism in Arabidopsis. (**A**) Enrichment analysis shows different epigenetic modification patterns across metabolic domains. The heatmap represents log2 fold change (Log2FC) of enrichment or depletion of an epigenetic mark associated with each domain relative to total metabolic genes. Epigenetic modifications are color-coded based on their effect on gene expression; red represents activating mark and blue represents repressive mark. Genes only mapped to unique metabolic domains were included in this analysis. Significant enrichment or depletion is based on Fisher's exact test, p-value < 0.05 and fold change >1.5. Gray cells represent no significant change. Black arrows indicate the patterns for trimethylation of lysine 27 of histone 3 (H3K27me3) and H3K18ac. (**B**) The co-occurrence of H3K27me3 and H3K18ac on total enzymatic genes in the genome and specialized metabolic genes.

The online version of this article includes the following figure supplement(s) for figure 1:

**Source data 1.** Source data used to generate *Figure 1A, B*.

**Figure supplement 1.** Correlation analysis of epigenetic marks based on their relative abundance on (**A**) total *Arabidopsis* metabolic genes, (**B**) non-specialized metabolic genes.

**Figure supplement 2.** Epigenetic marks associated with pathways involved in specialized metabolism including camalexin biosynthesis.

H3K18ac are associated with the same genes but in different cells, which may contribute to determining cell-type specificity of this pathway. To distinguish between the two possibilities, we examined in vivo co-localization of the two modifications using SeqChIP-qPCR, which requires a two-step, serial chromatin pull-down with antibodies against these two modifications. Though more than 10 genes were annotated to camalexin biosynthesis pathway in the Plant Metabolic Network database, we focused on the three experimentally validated genes that are essential in synthesizing camalexin,

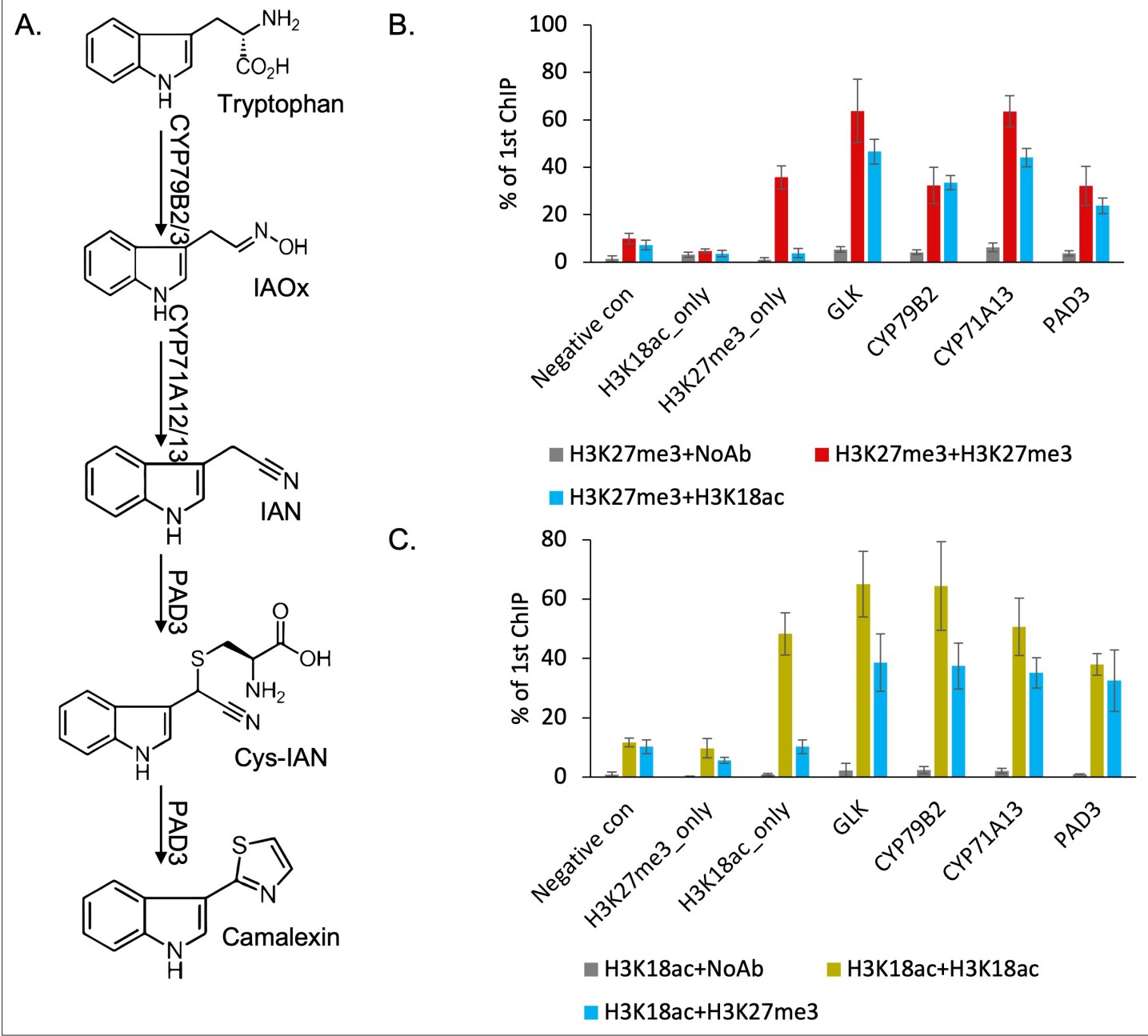

**Figure 2.** H3K27me3 and H3K18ac are co-localized on camalexin biosynthesis genes in planta to form a bivalent chromatin. (A) Camalexin biosynthesis pathway map and the essential genes that can produce camalexin. (B-C) Sequential chromatin immunoprecipitation (ChIP)-qPCR confirms the co-localization of H3K27me3 and H3K18ac. Altering the order of antibodies against H3K27me3 or H3K18ac in two rounds of pull-downs shows similar results in sequential ChIP-qPCR. Negative control represents a genomic region that has low abundance of both H3K27me3 and H3K18ac. H3K27me3_only represents a genomic region that has high abundance of H3K27me3 and low abundance of H3K18ac. H3K18ac_only represents a genomic region that has high abundance of H3K18ac and low abundance of H3K27me3. NoAb represents no antibody in the second pull-down. Error bars represent standard deviation of data from six biological replicates. The experiments were performed twice with different plant samples and data were combined in this analysis.

The online version of this article includes the following source data for figure 2:

**Source data 1.** Source data used to generate *Figure 2B,C*.

*CYP79B2*, *CYP71A13*, and *Phytoalexin Deficient 3* (*PAD3*) (**Klein et al., 2013**; **Schuhegger et al., 2006Figure 2A**). To assess the antibodies' specificity and efficiency, we included several negative and positive controls, including a transcription factor-encoding gene called *Golden-2-Like 1* (*GLK*) that was previously observed to be associated with both H3K27me3 and H3K18ac (**Luo et al., 2013**). All three camalexin biosynthesis genes showed significantly higher signals when pulled down with both antibodies against H3K27me3 and H3K18ac than without any antibody in the second pull-down (**Figure 2B**). We observed comparable pull-down efficiency for *GLK* between this study and the previous publication (**Zeng et al., 2019**). Altering the order of the two antibodies for H3K27me3 and H3K18ac in the pull-down showed similar patterns (**Figure 2C**). These results indicated that H3K27me3 and H3K18ac are co-localized at the camalexin biosynthesis genes in planta to form a bivalent chromatin.

The biological function of bivalent chromatins has long been hypothesized to poise gene expression for rapid activation upon signal perception. To date, this model has been tested only in mammalian stem cells and cancer cells (**Zeng et al., 2019**; **Zhang et al., 2021**; **Deng et al., 2013**), and no direct evidence is available in a whole organism context. As a first step to understand the role of the H3K27me3-H3K18ac bivalent chromatin on camalexin genes, we examined the transcriptional kinetics of camalexin biosynthesis genes under FLG22 induction in wild type and mutant lines with defective deposition of H3K27me3 or H3K18ac. We selected chromatin remodeler *pickle* (*pkl-1*) (**Carter et al., 2018**) and histone methyltransferase *curly leaf 28* (*clf28*) (**Doyle and Amasino, 2009**) to study the function of H3K27me3 on gene expression. The protein complex involved in establishing H3K18ac has not been well characterized in *Arabidopsis*, which limits the genetic resources that can be utilized to study the function of this epigenetic modification. The available mutant lines associated with establishing H3K18ac are *increased DNA methylation* 1 and 2 (*idm1* and *idm2*) (**Qian et al., 2014**; **Qian et al., 2012**). IDM1 is a histone acetyltransferase and IDM2 is a heat shock protein that functions in the same protein complex as IDM1 (**Qian et al., 2014**; **Qian et al., 2012**).

In wild-type plants, *CYP71A13* and *PAD3* were significantly induced within 30 min after FLG22 treatment and *CYP79B2* was significantly induced within 1 hr after the treatment (**Figure 3A-C**, **Supplementary file 3** to 5). In mutants with reduced deposition of H3K27me3 or H3K18ac, the induction patterns of the camalexin biosynthetic genes were disrupted. For *pkl-1* and *clf28* (reduced H3K27me3 marks), all three camalexin biosynthesis genes showed a significant induction of expression much faster than the wild type, within 5 min of FLG22 treatment, and the degree of induction was much higher than that in wild-type plants at 6 hr of the treatment (**Figure 3A** to C, **Supplementary file 3** to 5). In contrast, in the lines with reduced H3K18ac marks, *idm1* and *idm2*, genes were induced much later than the wild type. *CYP71A13* and *PAD3* were significantly induced within 1 hr after the treatment and *CYP79B2* was induced within 3 hr after the treatment (**Figure 3A** to C, **Supplementary file 3** to 5). To better understand the effect of genotype, FLG22 treatment, and duration of treatment on gene expression, we conducted a three-way ANOVA using raw transcripts of camalexin genes. Genotype, treatment, duration of treatment, and their interactions significantly contribute to the expression changes of *CYP71A13* and *PAD3* (**Supplementary file 5**). For *CYP79B2*, all experimental variables contributed significantly to its expression patterns except the genotype × treatment × duration of treatment (**Supplementary file 5**). These data generally support the idea that disrupting H3K27me3 and H3K18ac affects the temporal induction patterns of camalexin genes in response to FLG22 treatment.

To test whether the changes in gene expression would lead to changes in metabolite accumulation, we measured camalexin content using liquid chromatography–tandem mass spectrometry (LC-MS/MS). For wild-type plants, camalexin accumulation significantly increased 3 hr after FLG22 treatment and continued to increase 6 hr after the treatment (**Figure 3D**, **Supplementary file 6**). Plants with reduced H3K27me3 marks, *clf28 and pkl-1*, started to accumulate more camalexin earlier, 1 and 3 hr after FLG22 treatment, and continued to increase at later time points. On the other hand, for mutants with reduced H3K18ac marks, *idm1* and *idm2*, camalexin accumulated much later, at 6 hr after the treatment (**Figure 3D**, **Supplementary file 6**). We examined the effect of genotype, FLG22 treatment, and duration of the treatment on camalexin accumulation using three-way ANOVA (**Supplementary file 6**). Each of the three factors affected camalexin content significantly (p-values = 1.57E-05, 7.71E-09, 2.00E-16). However, the interaction between genotype and treatment and the three-way interaction, genotype × FLG22 × duration of treatment, were not significant (p-values =

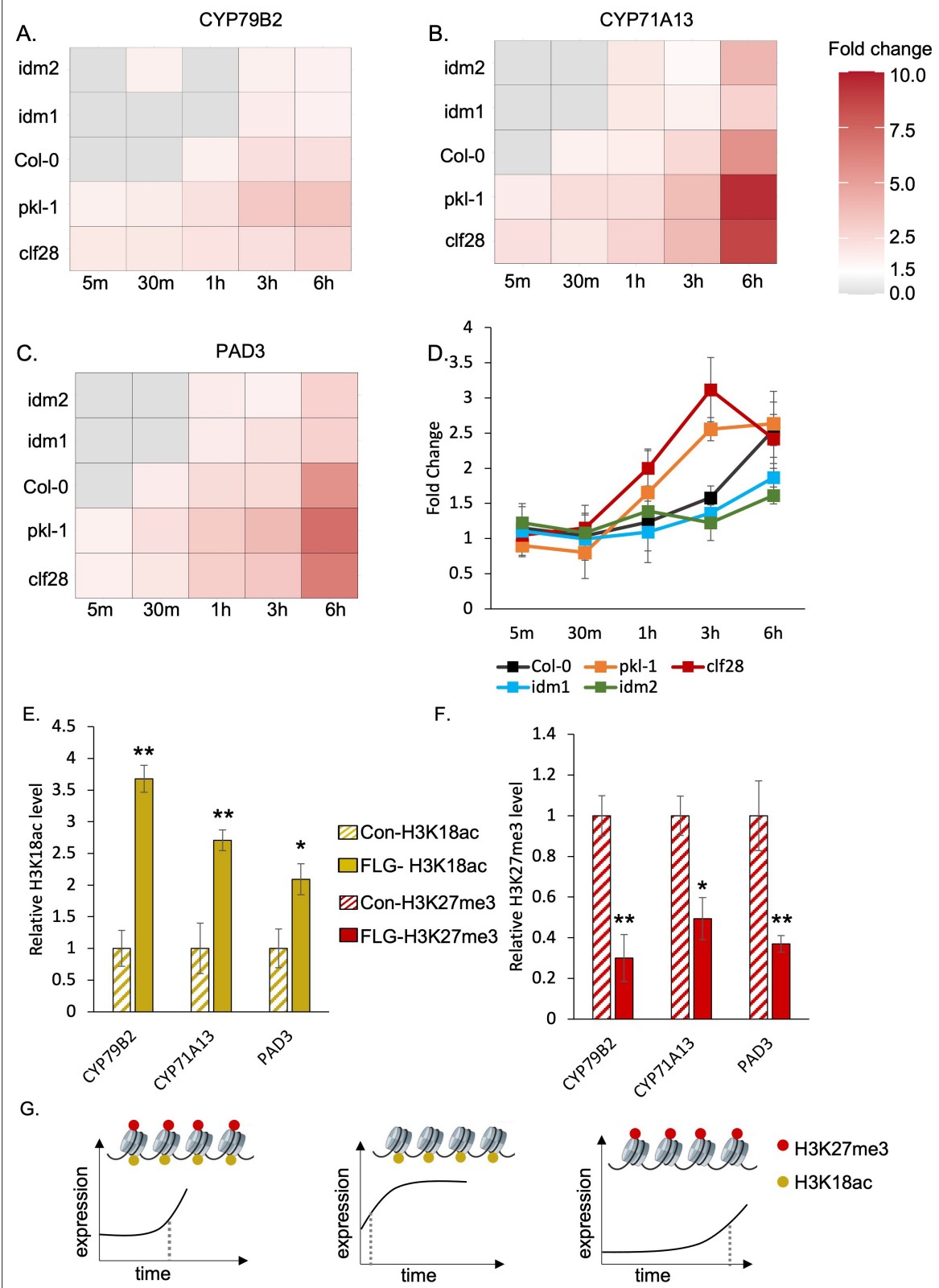

**Figure 3.** H3K27me3-H3K18ac bivalent chromatin controls the timing of gene induction and camalexin accumulation upon a pathogen signal. (**A-C**) Expression of the three essential genes in camalexin biosynthesis in response to flagellin 22 (FLG22) in the wild type (Col-0), H3K27me3-defective mutants *clf28* and *pkl-1*, and H3K18ac-defective mutants *idm1* and *idm2*. *ACT2* was used as a reference gene. Relative gene expression at each time point was reported in *Supplementary file 3* to 5. Fold change represents the induction level of camalexin biosynthesis genes in each genotype under

*Figure 3 continued on next page*

*Figure 3 continued*

FLG22 treatment relative to their corresponding untreated controls at each time point. (**D**) Camalexin accumulation in the wild type and mutants in response to FLG22. Camalexin content at each time point was reported in *Supplementary file 6*. Error bars represent standard deviation of data from six biological replicates of two experiments. (**E-F**) Abundance of epigenetic modifications at the genomic regions of camalexin biosynthesis genes with or without FLG22 treatment. Con-H3K18ac = chromatin was extracted from water-treated samples (Col-0) and pulled down by the antibody against H3K18ac. FLG-H3K18ac = chromatin was extracted from FLG22-treated samples (Col-0) and pulled down by the antibody against H3K18ac. Con-H3K27me3 = chromatin was extracted from water-treated samples (Col-0) and pulled down by the antibody against H3K27me3. FLG-H3K27me3 = chromatin was extracted from FLG22-treated samples (Col-0) and pulled down by the antibody against H3K27me3. Relative abundance of each modification was calculated by normalizing first to input, then to control plants with mock treatment. Two-week-old *Arabidopsis* seedlings of Col-0 were treated with 1 μM FLG22 or deionized water in these experiments. Stressed and control samples were collected 30 min after the treatment. Error bars represent standard deviation of data from six biological replicates of two experiments. * and ** represent p-values < 0.05 and 0.01, respectively, from Student's two-tailed t-test. (**G**) A model of the H3K27me3-H3K18ac bivalent chromatin that acts as a kairostat to regulate the temporal pattern of gene expression in response to external stimuli. Upon a pathogen signal, bivalent chromatin controls the timing of induction for camalexin biosynthetic genes, with H3K18ac expediting and H3K27me3 attenuating expression to hit the presumed temporal 'sweet spot'. Dashed lines represent the time of significant gene induction.

The online version of this article includes the following figure supplement(s) for figure 3:

**Source data 1.** Source data used to generate *Figure 3A-F*.

**Figure supplement 1.** Expression of *CYP79B2* in the wild type and mutants in response to flagellin 22 (FLG22).

**Figure supplement 2.** Expression of *CYP71A13* in the wild type and mutants in response to flagellin 22 (FLG22).

**Figure supplement 3.** Expression of *PHYTOALEXIN DEFICIENT 3* (*PAD3*) in the wild type and mutants in response to flagellin 22 (FLG22).

**Figure supplement 4.** Camalexin accumulation in the wild type and mutants in response to flagellin 22 (FLG22).

6.20E-02 and 9.50E-01). One possibility is that the kinetics of metabolite changes might be different from transcripts since significant accumulation of camalexin was detected at later time points. Another possibility is metabolite data being noisier than gene expression data. In summary, the transcriptional kinetics analysis suggests that H3K27me3-H3K18ac bivalent chromatin might control the timing of camalexin production in response to FLG22, which is partially corroborated with metabolite accumulation patterns.

We next asked whether FLG22 treatment changed the relative amount of H3K27me3 and H3K18ac marks on the camalexin biosynthetic genes. To address this question, we quantified the abundance of H3K27me3 and H3K18ac on the camalexin biosynthetic genes with or without FLG22 treatment using ChIP-qPCR. For each gene tested, primers were designed to cover a genomic region that included 1 kb upstream and the entire transcribed region (*Supplementary file 2*). In plants treated with FLG22, the abundance of H3K18ac was significantly higher in the gene regions of *CYP71A13*, *CYP79B2*, and *PAD3* compared to the mock-treated plants (*Figure 3E*). Moreover, the abundance of H3K27me3 modification was significantly lower in the regions of these genes in response to FLG22 (*Figure 3F*). These results suggest that the FLG22-induced changes of camalexin biosynthetic gene expression involve remodeling of the H3K27me3-H3K18ac bivalent chromatin. Taken together, this study demonstrates that a novel bivalent chromatin formed by H3K27me3 and H3K18ac controls the timing of induction of camalexin biosynthesis genes upon stress, with H3K18ac expediting and H3K27me3 attenuating expression to hit the presumed 'sweet spot' (*Figure 3G*). We have named this type of bivalent chromatin regulator a kairostat, inspired by the ancient Greek word 'kairos' meaning the right moment and 'stat' meaning regulating device.

## Discussion

Plant specialized metabolites are integral for plant-environmental interactions and defense against herbivores and pathogens. Camalexin is a prominent phytoalexin in *Arabidopsis* and genes associated with its biosynthesis are induced rapidly upon pathogen signals such as FLG22 (*Denoux et al., 2008*; *Stahl et al., 2016*; *Zipfel et al., 2004*). In this study, we found that H3K27me3 and H3K18ac co-localized on camalexin biosynthesis genes forming a previously uncharacterized form of bivalent chromatin. We examined the role of H3K27me3-H3K18ac bivalent chromatin on the induction of camalexin biosynthesis genes by combining publicly available epigenomics data with genetics, molecular biology, and biochemistry. In wild-type (Col-0) plants, camalexin biosynthesis genes were activated at 30 min to 1 hr of FLG22 treatment and continued to increase their expression at later time

points of the treatment. This is consistent with the previous study on induction kinetics of defense pathways, including camalexin, under FLG22 treatment (*Denoux et al., 2008*). In mutants with defective deposition of H3K27me3 or H3K18ac, the induction kinetics of camalexin biosynthesis genes and metabolite accumulation was accelerated or delayed, respectively.

Our results provide new evidence for how chemical defense mediated by camalexin may be regulated at the epigenetic level. However, we cannot rule out the possibility that other known mechanisms regulating camalexin genes may also affect transcriptional kinetics and metabolite accumulation. For example, H3K27me3 affects gene expression by altering chromatin accessibility to transcription factors (*Aranda et al., 2015*). Removing this repression mark may create a permissive environment and facilitate transcription factors, such as WRKY33, to bind to promoters of camalexin genes. At the translational level, camalexin biosynthesis genes can alter translational efficiency controlled by a highly enriched messenger RNA consensus sequence, R-motif, during pattern triggered immunity (*Xu et al., 2017*; *Yoo et al., 2020*). Additional studies are needed to unravel how different regulatory machineries work together to enable the rapid induction of camalexin genes upon stress signals.

In this study, we attempted to reveal the biological function of bivalent chromatin, which has long been hypothesized to poise developmental genes in embryonic stem cells for rapid activation upon a cell differentiation signal (*Zhang et al., 2009*). Only few studies directly tested this hypothesis and all of them focused on mammalian stem cells or cancer cells (*Deng et al., 2013*; *Lima-Fernandes et al., 2019*; *Jia et al., 2012*). For example, in embryonic stem cells, H3K27me3-H3K4me3 bivalent chromatin was associated with *HoxB4*, one of the key regulators controlling development. Inhibition of H3K4 methyltransferase, hSET1A, repressed the expression of *HoxB4* and blocked the differentiation of blood cells (*Deng et al., 2013*). In addition, H3K27me3-H3K4me3 bivalent chromatin was observed in cancer-initiating cells. Disruption of the bivalent state through inhibition of the H3K27 methyltransferase EZH2 inhibited the self-renewal of cancer cells through de-repression of a key canonical marker of normal colonocyte differentiation, named Indian Hedgehog (*Lima-Fernandes et al., 2019*). Despite this knowledge, the role of bivalent chromatin on regulating metabolism remains unknown. In this study, we reported the co-localization of H3K27me3 and H3K18ac on camalexin biosynthesis genes and functionally examined the role of H3K27me3-H3K18ac bivalent chromatin in *Arabidopsis*. The results suggest that H3K27me3-H3K18ac bivalent chromatin maintains the proper timing of camalexin gene induction upon a pathogen signal.

To fully elucidate how bivalent chromatin integrates environmental cues to regulate gene expression, it would be critical to understand how bivalent chromatin is established and maintained and how different epigenetic marks function cooperatively to control gene expression. To achieve these goals, it is required to characterize all the proteins involved in establishing different epigenetic marks. Mechanisms that enable histone modification protein complexes to alter the abundance of each mark upon stress signals also remains to be elucidated. A recent study showed that NF-Y transcription factors can interact with histone modification protein REF6 to reduce H3K27me3 at the *SOC1* locus (*Hou et al., 2014*). It would be interesting to examine whether transcription factors that directly bind to promoters of camalexin biosynthesis genes interact with proteins that can deposit or remove both epigenetic marks forming bivalent chromatin upon stress signals. The H3K27me3-H3K18ac bivalent chromatin and its role on the camalexin biosynthesis genes described here provide the molecular handle with which to open these lines of investigations in the future. With these knowledge gaps filled, kairostats can be engineered to function as a regulatory cassette to control the expression of target genes under specific tissues and conditions. This capability will provide novel routes to re-program biological processes involved in development and stress response, which have far-reaching implications in agriculture, engineering, and medicine.

## Materials and methods
### Annotation of metabolic genes, pathways, and domains

*Arabidopsis* metabolic enzymes and pathways were extracted from Plant Metabolic Network (http://www.plantcyc.org/). Enzymes were annotated based on the Ensemble Enzyme Prediction Pipeline (E2P2) v.3.0 (*Schläpfer et al., 2017*; *Chae et al., 2014*), metabolic pathways were predicted by Pathway Tools software v.18.5 (*Karp et al., 2011*), which were further validated by a process called Semi-Automated Validation and Integration (SAVI) v.3.02 (*Zhang et al., 2010*). In *A. thaliana*, 5451

genes were predicted to encode enzymes that catalyze 3585 reactions in small molecule metabolism, which were assigned to 627 pathways based on prediction and manual curation in the release of PMN 13.0 (*Schläpfer et al., 2017*). These pathways were further grouped into 13 metabolic domains based on the types of metabolites that they potentially synthesize (*Schläpfer et al., 2017*). The metabolic domain(s) assigned to a given pathway were transferred to the reactions, EC numbers, and proteins/genes associated with that pathway. To improve the statistical power, we included only those pathways that contained more than 10 genes when we performed epigenetic modification enrichment analysis on individual pathways in specialized metabolism (*Sham and Purcell, 2014*).

## Epigenomic enrichment analysis

The 16 epigenomic profiles were downloaded from *Wang et al., 2015*, where previously published ChIP-seq and ChIP-chip data for all epigenetic marks were re-mapped to *A. thaliana* genome TAIR v.10 (*Wang et al., 2015*). Briefly, for each ChIP-seq profile, the signals were binned and ranked in the genome (*Wang et al., 2015*). We used the ranked epigenetic modification signals in the following analyses.

To identify the predominant epigenetic modifications associated with metabolic domains and pathways, we first mapped the ranked epigenetic modification signals to each gene, including the transcribed region, 1 kb upstream, and 500 bp downstream regions. Then, we calculated the average epigenetic modification density for each domain or pathway by taking the sum of the ranked modification signals observed within all the genes in that domain or pathway and normalizing it to the total length of all genes. To identify enriched marks within metabolic domains or pathways, we compared the density of each epigenetic modification within that domain or pathway to the background signal. We defined the background signal as the average density of each epigenetic modification by taking the sum of the epigenetic modification signals observed within total metabolic gene regions and normalizing it to the length of these genes.

The statistical significance of enrichment or depletion of epigenetic marks per domain or pathway was determined by fold change of at least 1.5 and Fisher's exact test p-value, followed by a post hoc adjustment using false discovery rate (FDR) with the threshold of 0.01. The heatmaps were generated with gplots v.3 (*Warnes, 2009*) and ggplot2 v.3.1 (*Wickham, 2016*) packages in RStudio.

## Plant materials and growth conditions

Seeds of *A. thaliana* accession Columbia (Col-0) (wild type) and previously characterized mutant lines *clf28* (SALK_139371), *pkl-1* (SALK_010693), *idm1* (SALK_062999), and *idm2* (SALK_138229) were used in this study. Seeds of these mutants were obtained from the *Arabidopsis* Biological Resource Center. In all the experiments described in this study, seeds were stratified at 4 °C for 3 days before germination and grown in 0.5 × Murashige and Skoog (MS) medium containing 1 % sucrose and 0.8 % agar at 23 °C under long days (16 hr light/8 hr dark) with controlled light at 100 µmol photosynthetic photons/m$^2$ s (PPFD). Two-week-old seedlings were harvested for the ChIP-qPCR, SeqChIP-PCR, transcriptional kinetics, and camalexin content measurement experiments.

## ChIP-qPCR and SeqChIP-PCR

Two-week-old *Arabidopsis* seedlings (Col-0) were used in ChIP and SeqChIP (sequential ChIP) experiments. To obtain sufficient chromatin for immunoprecipitation, 200 seedlings (about 2 g) were pooled for each ChIP reaction. Plant tissues were fixed with crosslinking buffer containing 1 % formaldehyde for 20 min by vacuum infiltration at room temperature and quenched with glycine (final concentration 100 mM) for 5 min. After grinding, the cross-linked chromatin was isolated using ice-cold nuclear lysis buffer (50 mM HEPES at pH 7.5, 150 mM NaCl, 1 mM EDTA, 1 % Triton X-100, 0.1 % Na deoxycholate, 0.1 % SDS, and protease inhibitor) and disrupted by sonication to yield DNA fragments of 200–600 bp (*Zhu et al., 2016*; *Saleh et al., 2008*).

To examine the abundance of H3K27me3 and H3K18ac 30 min after 1 µM FLG22 treatment, ChIP-qPCR was used to quantify their abundance throughout the genomic region and 1 kb upstream of the three essential camalexin biosynthesis genes. After chromatin crosslinking and shearing, DNA/protein complexes were equally divided and immunoprecipitated with antibodies against H3K27me3 (Millipore, Cat No 07–449), H3K18ac (Abcam, Cat No ab1191), or IgG as antibody control (Millipore, Cat No 12–370) by referring the ChIP protocol established using *Arabidopsis* tissues (*Zhu et al., 2016*;

*Saleh et al., 2008*). DNA was purified using QIAquick PCR Purification Kit (Qiagen, Cat No 28104). Quantitative real-time PCR was performed using LightCycler 480 System (Roche) with the SensiFAST Sybr No-Rox mix (Bioline, Cat No BIO98020) from Bioline. Three biological replicates were used in each experiment. The ChIP experiments were performed twice with different *Arabidopsis* tissues and revealed similar results. Primers were designed to cover the 1 kb upstream of the transcription start site and the entire transcribed region (*Supplementary file 2*).

SeqChIP experiments were performed to examine in planta co-localization of H3K27me3 and H3K18ac on camalexin biosynthesis genes. Chromatin crosslinking and shearing followed the protocol described above for ChIP-qPCR. For each ChIP/re-ChIP assay, 20 µg DNA/protein complexes were immunoprecipitated with anti-H3K27me3 (Millipore, Cat No 07–449) and anti-H3K18ac (Abcam, Cat No ab1191) antibodies using the Re-ChIP-IT kit (Active Motif, Cat No 53016). Quantitative real-time PCR was performed using LightCycler 480 System (Roche) with the SensiFAST Sybr No-Rox mix (Bioline, Cat No BIO98020) from Bioline. To represent final results after two-step chromatin immunoprecipitation, the pull-down efficiency was calculated as the percent of first ChIP using the equation: 1st ChIP = $2^{\wedge(Cq\_2nd\ ChIP - Cq\_1st\ ChIP)}$. Three biological replicates were used in each experiment. The SeqChIP was performed twice using different *Arabidopsis* tissues and altering the order of antibodies in two rounds of pull-down showed similar results. Primer sequences used in this assay are provided in *Supplementary file 3*.

## Transcriptional kinetics analysis

Two-week-old seedlings grown in MS agar media as described in the Plant materials and growth conditions section were used to study transcriptional kinetics. FLG22, a short peptide of 22 amino acids (sequence: QRLSTGSRINSAKDDAAGLQIA), was synthesized at Stanford Protein and Nucleic Acid (PAN) Facility. To induce the expression of camalexin biosynthesis genes, seedlings were sprayed with 1 µM FLG22 dissolved in deionized (DI) water. Control samples were sprayed with the same volume of DI water. FLG22 and water-treated tissues were sampled at 5 min, 30 min, 1 hr, 3 hr, and 6 hr after treatment and frozen immediately in liquid nitrogen.

After grinding in liquid nitrogen, RNA was extracted using RNeasy Plant Mini Kit (Qiagen, Cat No 74904) and 2 µg RNA was used for cDNA synthesis using SuperScript First-Strand Synthesis System (Thermo Fisher, Cat No 18080051). Quantitative real-time PCR was performed using LightCycler 480 System (Roche) with the SensiFAST Sybr No-Rox mix (Bioline, Cat No BIO98020) from Bioline. Primers used in this experiment are summarized in *Supplementary file 4*. A housekeeping gene, Actin 2 (AT3G18780), was used as the reference gene in the qPCR experiments. Three biological replicates were used in each experiment by polling 10 seedlings per replicate. The transcriptional kinetics assays were performed twice and showed similar results.

## Camalexin content quantification using LC-MS

Two-week-old seedlings of Col-0, *pkl-1*, *clf28*, *idm1*, and i*dm2* plants were used to measure camalexin accumulation. Tissues were treated with FLG22 or DI water as described in the "Transcriptional kinetics analysis" section. Metabolites were extracted using a 75 % methanol solution with 0.1 % formic acid. For each sample, 150 mg of ground plant tissue was dissolved in 200 µL extraction solution. The extracted samples were vortexed at room temperature for 40 min and centrifuged at 32,000 *g* for 10 min to pellet plant debris. The supernatant was analyzed by an Agilent 1260 Infinity Binary HPLC paired with an Agilent 6420 Triple Quadrupole LC-MS/MS, with a reversed-phase column (Agilent EclipsePlus C18, 2.1 × 50 mm, 1.8 µm), water with 0.1 % formic acid as solvent A and acetonitrile with 0.1 % formic acid as solvent B, at a constant flow rate of 0.4 mL/min and an injection volume of 10 µL. The following gradient was used for metabolite separation: 0–0.5 min, 10–50% B; 0.5–5.5 min, 50–98% B; 5.5–6 min, 98 % B; 6.00–6.01 min, 98–10% B; 2 min post-time equilibration with 10 % B. The LC eluent was subjected to MS for 0.01–6.01 min with the ESI source in positive mode, gas temperature at 350 °C, gas flow rate at 11 L/min, and nebulizer pressure at 40 psi. To detect camalexin, we set the multiple reaction monitoring (MRM) ion transition to be 201.26 → 59.2, fragmentor at 135 V, and collision energy at 40 V. The MRM ion transitions in this work were derived from product ion scan with precursor ion set at 201.26 and the most abundant product ion was chosen for MRM transition quantification. LC-MS/MS data files were analyzed using Agilent MassHunter Workstation software v.B.06.00. Metabolite content was quantified by integrating peak area under the ion count

curve. The ion counts were calibrated to camalexin chemical standard (Cat Num Sigma SML1016) and converted to measurements of titer in molar concentrations (nM). Three biological replicates were used in each experiment by polling 50 seedlings per replicate. The experiments were conducted twice with different plant samples and showed similar results.

## Acknowledgements

We thank Jennifer Brophy and Flavia Bossi for their critical feedback on this work. We acknowledge Jia-Ying Zhu and Yuchun Hsiao for help with ChIP-qPCR experiments and Hye-In Nam for help with growing plants. Funding: This work was supported in part by Carnegie Institution for Science Endowment and grants from the National Science Foundation (IOS-1546838, IOS-1026003), the U.S. Department of Energy, Office of Science, Office of Biological and Environmental Research, Genomic Science Program grant nos. DE-SC0018277, DE-SC0008769, and DE-SC0020366, and the National Institutes of Health (1U01GM110699-01A1). This work was done on the ancestral land of the Muwekma Ohlone Tribe, which was and continues to be of great importance to the Ohlone people.

## Additional information

### Funding

| Funder | Grant reference number | Author |
|---|---|---|
| National Science Foundation | IOS-1546838 and IOS-1026003 | Kangmei Zhao<br>Benjamin Jin<br>Seung Yon Rhee |
| National Institute for Health Research | 1U01GM110699-01A1 | Kangmei Zhao<br>Deze Kong<br>Christina D Smolke<br>Seung Yon Rhee |
| U.S. Department of Energy | DE-SC0018277 | Kangmei Zhao<br>Seung Yon Rhee |
| Carnegie Institution for Science | Endowment | Kangmei Zhao<br>Benjamin Jin<br>Seung Yon Rhee |
| U.S. Department of Energy | DE-SC0008769 | Seung Yon Rhee |
| U.S. Department of Energy | DE-SC0020366 | Seung Yon Rhee |

The funders had no role in study design, data collection and interpretation, or the decision to submit the work for publication.

### Author contributions

Kangmei Zhao, Conceptualization, Data curation, Formal analysis, Investigation, Methodology, Writing - original draft, Writing - review and editing; Deze Kong, Formal analysis, Investigation, Methodology, Writing - original draft, Writing - review and editing; Benjamin Jin, Investigation; Christina D Smolke, Supervision, Writing - review and editing; Seung Yon Rhee, Funding acquisition, Supervision, Writing - original draft, Writing - review and editing

### Author ORCIDs

Kangmei Zhao http://orcid.org/0000-0003-1472-4993
Deze Kong http://orcid.org/0000-0003-4334-5888
Christina D Smolke http://orcid.org/0000-0002-5449-8495
Seung Yon Rhee http://orcid.org/0000-0002-7572-4762

### Decision letter and Author response

Decision letter https://doi.org/10.7554/eLife.69508.sa1
Author response https://doi.org/10.7554/eLife.69508.sa2

# Additional files

## Supplementary files

- Supplementary file 1. Log2 fold change in the enrichment analysis to identify specialized metabolic pathways associated with both trimethylation of lysine 27 of histone 3 (H3K27me3) and H3K18ac.
- Supplementary file 2. Primers used to quantify the abundance of trimethylation of lysine 27 of histone 3 (H3K27me3) and H3K18ac in the genomic regions of camalexin biosynthesis genes using chromatin immunoprecipitation (ChIP)-qPCR in wild-type and mutant plants with or without flagellin 22 (FLG22) treatment.
- Supplementary file 3. Primers used in the sequential chromatin immunoprecipitation (ChIP)-qPCR experiment to examine the co-localization of trimethylation of lysine 27 of histone 3 (H3K27me3) and H3K18ac within camalexin biosynthesis genes.
- Supplementary file 4. Primers used to examine the expression of camalexin biosynthesis genes under flagellin 22 (FLG22) induction using qPCR.
- Supplementary file 5. The effect of genotype, flagellin 22 (FLG22) treatment, and time point on the expression change of camalexin genes.
- Supplementary file 6. The effect of genotype, flagellin 22 (FLG22) treatment, and time on camalexin accumulation.
- Transparent reporting form

## Data availability

The data that support the findings of this study are available within the article and its supplementary information files. Arabidopsis metabolic genes and pathways are available at Plant Metabolic Network (https://plantcyc.org/).

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
