## [Decision Letter]

**Acceptance summary:**

This manuscript provides evidence that chromatin with activating and repressing histone modifications at the same location enables the rapid activation of genes required for the biosynthesis of camalexin, a plant defence compound, upon detection of a pathogen signal. The data show that both H3K27me3 and H3K18ac are required to determine the timing of gene expression.

**Decision letter after peer review:**

[Editors’ note: the authors submitted for reconsideration following the decision after peer review. What follows is the decision letter after the first round of review.]

Thank you for submitting your work entitled "A Bivalent Chromatin Controls Timing of Expression of Camalexin Biosynthesis Genes in Response to a Pathogen Signal in Arabidopsis" for consideration by *eLife*. Your article has been reviewed by 3 peer reviewers, and the evaluation has been overseen by a Reviewing Editor and a Senior Editor. The reviewers have opted to remain anonymous.

Comments to the Authors:

We are sorry to say that, after consultation with the reviewers, we have decided that your work will not be considered further for publication by *eLife*.

In this manuscript, the authors present data showing that changes to H3K27me3/H3K18ac marks help regulate the main Arabidopsis phytoalexin. While the chromatin states of these specific genes have not previously been reported on directly, this form of regulation is known to occur in plants and H3K27me3 and H3K18ac have previously been shown to colocalize in a large number of genes in the Arabidopsis genome. Further, H3K27me3 marked genes are known to be targets of Polycomb Repressive Complex 2 (PRC2) and, therefore, the changes in the regulation of such genes in H3K27me3 defective mutants are as expected. The term bivalent chromatin has been misused in this manuscript to describe overlapping chromatin marks, which has led to the formation of confusing and incorrect hypotheses. In addition, concerns were raised about the pathway enrichment analysis and if this allows extension to specialized metabolism pathways in general. There is a potential for over-counting pathways when a metabolic pathway is inaccurately described by hierarchical database design. The phenomenon appears limited to fewer pathways than suggested and some statistical analysis are needed to support the broader claims. Finally, the changes to metabolite levels should be interpreted with reference to previously published data showing that post-translational mechanisms play a role in their regulation.

*Reviewer #1 (Recommendations for the authors):*

The focus of the manuscript is to investigate changes in chromatin state. The series of experiments to identify and examine changes in chromatin marks in response to flgg22 treatments investigate clearly defined hypotheses and the results are clear and accurately interpreted. Where the manuscript needs to be improved is the wider interpretation of the results in relation to changes in overall metabolite levels. The finding of bivalent chromatin and its role in gene regulation in response to biotic stress is interesting. However, the data is presented within a very narrow focus and interpretation of the findings within the wider background of literature about plant metabolic responses to pathogen infection is too limited. The manuscript would be improved by the inclusion of introductory sentences and a Discussion section focussing on how these new discoveries fit into the wider understanding of camalexin expression in response to pathogen infection. Though clearly chromatin state plays a role, it is one part of a multi-layered response that also includes changes in the rate of translation calcium-dependent phosphorylation of WRKY33 to enable to initiate transcription.

*Reviewer #2 (Recommendations for the authors):*

Technical suggestions.

In a number of cases, the pathway enrichment analysis gives the impression that there are a number of different pathways being linked to bivalent chromatin. However, it isn't clear if these are really different pathways or a construct of the categorization system. For example, in Figure 4B, Quercetin and Rutin biosynthesis each have two genes but Rutin is simply a Quercetin glycoside and so it is likely that these two genes are the same and this isn't two different pathways. There are other groupings like (Simple coumarins, Scopolin and phenylpropanoid; and Glucosinolate, indole glucosinolate and camalexin) where the separate genes are not really finding multiple different pathways but simply the same genes are finding different hierarchical categorizations. There is a similar complexity with at least six glucosinolate from Xmethionine treated as independent pathways as well as aliphatic glucosinolate, an overarching category encompassing these other terms. Yet these pathways are an artifact of glucosinolates involving a cyclic step that is determined by one enzyme (MAM1/2) adding a carbon and a second enzyme (CYP79F1/F2) having specificity for the chain length. Other than those two steps the rest of genes in these pathways. The complexity is that this cyclic structure can't be shown in a hierarchical construction. Thus to make the programming work they are broken into different "pathways". In general this makes it unclear if the specialized pathway enrichment analysis is being driven by an actual enrichment for all of specialized metabolism or did the identified genes just happen to track with these artificial pathways that can create the impression of all specialized metabolism. The key to this is to dissect how much the enrichment is driven by the same gene driving the identification of multiple pathways. Right now, it seems that this is not an enrichment with 23 independent specialized metabolism pathways but instead being driven by the issues of hierarchical programming and non-linear pathways in the original databases.

I should be clear that the above concern does not affect the specific camalexin molecular analysis, it simply suggests that the generalizations to all of specialized metabolism might be unsupported.

There are some missing statistical analysis. In Figure 3A-C, there should be some statistical analysis of the transcsript data as camalexin transcripts can easily have 3-4x biological variation so simply showing the fold-change doesn't provide information about what differences are meaningful in this pathway. Similarly, in Figure 3D, there should be some statistical analysis of the camalexin to test if there are time x genotype dependent effects on the metabolite. The standard error is not a statistical test. Similarly, the other replicates of this experiment should be combined into this analysis. Camalexin metabolites are fairly noisy and all the data should be combined and analyzed to show how reproducible the effects are.

I don't think the literature provides sufficient resolution to say that only CYP79B2 is involved in camalexin as is proposed in Figure 2. CYP79B2 KOs an induce camalexin suggesting that CYP79B3 is also involved. Similarly they CYP71A12/A13 specificity is still unresolved. I understand the focus when doing the CHIP but the genomic data should have information about these other genes and could be provided in the supplemental.

The metabolomics in response to FLG22 was intended to show that the gene expression changes were indicative of metabolite changes. Unfortunately, the MS finds largely unknown compounds which could be all related to camalexin synthesis or other CYP79B2 dependent metabolites. As such, it doesn't really support the broader argument about all specialized metabolites. There is also the chance that the enhanced response in clf28 could be due to lower basal levels. Right now with FC levels shown it isn't clear if this is an enhanced response or enhanced repression. It would help to show absolute levels and to conduct factorial analysis testing for genotype and genotype x treatment effects. This is a similar concern for the microarray analysis as well.

Editorial Suggestions

I'm not totally sure what the main point of the second paragraph of the discussion is conveying. This paragraph starts with ideas about bivalent chromatin in development and then ends with specific details about bivalent chromatin in camalexin. I feel there is a big point in this paragraph but as written I can't quite grasp it. *eLife* allows more space then is used and it would help the general reader to expand this section to let the big point come through clearer. I think it is about bivalent chromatin being a trigger switch when something must go from absolute off to absolute on and do so very quickly.

Related to the general idea of bivalent chromatin and camalexin, there are some complications around this pathway that aren't being integrated into the discussion. It doesn't change the results of this work but it does lay out a more complicated view then is being proposed. For example, the PAD mutants from Jane Glazebrook were isolated based on the accumulation of camalexin. A result of this work was the fact that the regulatory mutants (e.g. PAD4, etc) were highly specific to the stimulus being applied and often did not affect camalexin regulation in response to other stimuli. There is also work showing that the camalexin pathway has post-transcriptional regulation where the transcripts can accumulate to a high level with no metabolite accumulation. Finally, the FLG22 ability to induce camalexin is age dependent and is not seen in older soil grown plants based on the MAPK studies. Thus, the upstream and downstream regulatory processes around camalexin are significantly more complicated than the intro and discussion lays out by describing a couple of TFs. It would help to expand the discussion and introduction to convey this full complexity that is present in the established literature as it is critical to place the bivalent chromatin into the full context of camalexin regulation.

*Reviewer #3 (Recommendations for the authors):*

The genes being studied are targeted by the PRC2 complex and this is why they have H3K27me3. Past studies have discovered that H3K18ac and H2A.Z colocalize with H3K27me3 (Luo C, 2012). Yet, there is no mention of H2A.Z throughout this study? Would this now make this chromatin state 'trivalent' chromatin? This issue specifically highlights the incorrect premise of this study, that there is a "bivalent chromatin" at these metabolic genes. They, like many other H3K27me3 marked genes, are targets of PRC2 and silenced until proper environmental/development cues are perceived. This fits with the highly specialized need for these metabolic genes in unique environments.

There is no evidence presented that H3K18ac is an 'activating" mark. This should not be confused with other histone acetylation marks. Based on what was previously described by Luo C et al., 2012, it is a repressive mark given it's colocalization with H3K27me3.

How do you distinguish direct versus indirect effects when using pkl/clf/idm1 mutants? These genes are required for general gene regulation. For example, without clf, no H3K27me3 occurs leading to up-regulation of all PRC2 targeted genes?

I have major concerns that IDM1 is the HAT that acetylates H3K18 in gene bodies. IDM1 specifically binds to methylated DNA, which does not colocalize with H3K27me3 in Arabidopsis (see also Figure 1a). This would also explain the seemingly weak signals from all experiments using idm1. ChIP-seq data for IDM1 exists (https://www.ncbi.nlm.nih.gov/pmc/articles/PMC3575687/) and should be used to show it colocalizes specifically to the genes that have H3K27me3 versus other types of genes.

The term bivalent chromatin should be more carefully used in this study. Bivalent chromatin specifically refers to H3K4me3/H3K27me3. It should not be used to described H3K27me3/H3K18ac. There are numerous examples where bivalent chromatin is referring to the classic definition (H3K4me3/H3K27me3) to set up rationale for proposed experiments on H3K27me3/H3K18ac. This leads to incorrect hypotheses and is confusing to the reader.

---

## [Author Response]

[Editors’ note: The authors appealed the original decision. What follows is the authors’ response to the first round of review.]

Reviewer #1 (Recommendations for the authors):The focus of the manuscript is to investigate changes in chromatin state. The series of experiments to identify and examine changes in chromatin marks in response to flgg22 treatments investigate clearly defined hypotheses and the results are clear and accurately interpreted. Where the manuscript needs to be improved is the wider interpretation of the results in relation to changes in overall metabolite levels. The finding of bivalent chromatin and its role in gene regulation in response to biotic stress is interesting. However, the data is presented within a very narrow focus and interpretation of the findings within the wider background of literature about plant metabolic responses to pathogen infection is too limited. The manuscript would be improved by the inclusion of introductory sentences and a Discussion section focussing on how these new discoveries fit into the wider understanding of camalexin expression in response to pathogen infection. Though clearly chromatin state plays a role, it is one part of a multi-layered response that also includes changes in the rate of translation calcium-dependent phosphorylation of WRKY33 to enable to initiate transcription.

We thank the reviewer for pointing this out and helping us improve the manuscript by providing a more holistic context of camalexin regulation. We revised Introduction and Discussion to include current understanding of post-transcriptional regulation on camalexin pathway and interpret the results in a more holistic view. Revisions in the Introduction:

“Previous studies revealed complex transcriptional and translational control of camalexin biosynthesis genes^8, 9, 10, 11, 12, 13^. At the transcriptional regulation level, transcription factors from the MYB family, including MYB34, MYB51 and MYB122 promote camalexin biosynthesis in response to *P. syringae* infection^9, 12^. WRKY33 functions as an activator and directly binds to the promoters of camalexin biosynthesis genes^11^. Besides transcription factors, CALCIUM-DEPENDENT PROTEIN KINASE (CPK)5/6 and MAPK3/6 can phosphorylate WRKY33 to enhance promoter binding and transactivation^10, 13^. At the translational level, ribosome footprinting showed that genes involved in camalexin biosynthesis, CYP79B2 and CYP79B3, also increased translational efficiency under pattern triggered immunity^14, 15^. Despite the rich knowledge of these upstream regulators of the camalexin biosynthetic pathway, it remains unknown how the rapid induction upon a pathogen signal is enabled”.

In Discussion, we added:

“Our results provide new evidence for how chemical defense mediated by camalexin may be regulated at the epigenetic level. However, we cannot rule out the possibility that other known mechanisms regulating camalexin genes may also affect the transcription kinetics and metabolite accumulation. For example, H3K27me3 affects gene expression by altering chromatin accessibility to transcription factors ^21^. Removing this repression mark may create a permissive environment and facilitate transcription factors, such as WRKY33, to bind to promoters of camalexin genes. At the translational level, camalexin biosynthesis genes can alter translational efficiency controlled by a highly enriched messenger RNA consensus sequence, R-motif, during pattern triggered immunity ^14, 15^. Additional studies are needed to unravel how different regulatory machineries work together to enable the rapid induction of camalexin genes upon stress signals.”

Reviewer #2 (Recommendations for the authors):Technical suggestions.In a number of cases, the pathway enrichment analysis gives the impression that there are a number of different pathways being linked to bivalent chromatin. However, it isn't clear if these are really different pathways or a construct of the categorization system. For example, in Figure 4B, Quercetin and Rutin biosynthesis each have two genes but Rutin is simply a Quercetin glycoside and so it is likely that these two genes are the same and this isn't two different pathways. There are other groupings like (Simple coumarins, Scopolin and phenylpropanoid; and Glucosinolate, indole glucosinolate and camalexin) where the separate genes are not really finding multiple different pathways but simply the same genes are finding different hierarchical categorizations. There is a similar complexity with at least six glucosinolate from Xmethionine treated as independent pathways as well as aliphatic glucosinolate, an overarching category encompassing these other terms. Yet these pathways are an artifact of glucosinolates involving a cyclic step that is determined by one enzyme (MAM1/2) adding a carbon and a second enzyme (CYP79F1/F2) having specificity for the chain length. Other than those two steps the rest of genes in these pathways. The complexity is that this cyclic structure can't be shown in a hierarchical construction. Thus to make the programming work they are broken into different "pathways". In general this makes it unclear if the specialized pathway enrichment analysis is being driven by an actual enrichment for all of specialized metabolism or did the identified genes just happen to track with these artificial pathways that can create the impression of all specialized metabolism. The key to this is to dissect how much the enrichment is driven by the same gene driving the identification of multiple pathways. Right now, it seems that this is not an enrichment with 23 independent specialized metabolism pathways but instead being driven by the issues of hierarchical programming and non-linear pathways in the original databases.I should be clear that the above concern does not affect the specific camalexin molecular analysis, it simply suggests that the generalizations to all of specialized metabolism might be unsupported.

In general, we really appreciate the technical comments from reviewer #2 and believe our manuscript was improved by addressing their comments. This comment makes two points: (1) redundancy of reaction mapping to pathways in specialized metabolism and potential inflation of pathway enrichment in specialized metabolism and (2) the claim of the bivalent chromatin being generalized to specialized metabolism.

The reviewer brought up an important point regarding the gene redundancy in pathway enrichment analysis. This made us re-evaluate the pathway organization in Plant Metabolic Network databases. We are now exploring different strategies to organize pathways to address the shared reaction issue for enrichment analysis, which will require a substantial effort and we feel is out of the scope of this manuscript. Nonetheless, we feel this is an important caveat of the pathway enrichment analyses of metabolic pathway databases.

To address the comment more directly, we reported the number of metabolic genes marked by both marks by stating:

“To determine how prevalent the H3K27me3-H3K18ac association was for specialized metabolic genes, we counted the number of genes marked by both modifications. We found that 37% (324 out of 887) of specialized metabolic genes were marked by both H3K27me3 and H3K81ac (Figure 1B).”

Enrichment analyses show increased propensity for the occurrence of a phenomenon to a specific group (e.g. specialized metabolism vs total metabolism), but does not imply the occurrence to all members of the enriched group (Reimand et al., (2019) Nature Protocol). To see which additional pathways might be regulated similarly as camalexin pathway, we used RNA-seq and untargeted metabolomics to measure gene expression and metabolites change in different genotypes under FLG22 treatment. Several pathways were identified using enrichment analysis with metabolic genes showing similar gene expression patterns as camalexin genes. However, we feel that these are preliminary at best and divert attention away from the major discovery of a new type of bivalent chromatin and its potential biological role on camalexin biosynthesis pathway. Thus, we removed sections reporting these results in the manuscript.

There are some missing statistical analysis. In Figure 3A-C, there should be some statistical analysis of the transcsript data as camalexin transcripts can easily have 3-4x biological variation so simply showing the fold-change doesn't provide information about what differences are meaningful in this pathway. Similarly, in Figure 3D, there should be some statistical analysis of the camalexin to test if there are time x genotype dependent effects on the metabolite. The standard error is not a statistical test. Similarly, the other replicates of this experiment should be combined into this analysis. Camalexin metabolites are fairly noisy and all the data should be combined and analyzed to show how reproducible the effects are.

We wanted to show how the transcriptional kinetics of camalexin genes changed upon FLG22 treatment at different time points and we feel that fold change reflects the change of gene expression best. To better understand the effect of genotype, FLG22 treatment, and time points on gene expression, we conducted three-way ANOVA using raw transcripts of camalexin genes. Supplementary File 5 summarized whether each variable or interactions between variables significantly contributed to the variance among groups. The results showed that genotype, treatment, time point, and their pairwise interaction significantly contributes to the expression change of camalexin genes. The three-way interaction term Genotype: Treatment: Time was significant for CYP71A13 and PAD3 but not for CYP79B2.

To better reflect the effect of genotype and treatment on camalexin gene expression, we generated boxplots for these genes and compared the difference between mock and FLG22 treated samples using two-way ANOVA followed by Dunnett’s test. The results showed that camalexin genes were induced 5min after FLG22 treatment in mutant plants defective of H3K27me3. In mutant defective of H3K18ac, the significant induction of camalexin genes was detected at 1h after FLG22 treatment.

For camalexin accumulation, as the reviewer suggested, we examined the effect of genotype, FLG22 treatment, and time point on camalexin gene expression using three-way ANOVA by combining all technical and biological replicates from two independent experiments. In Supplementary file 6 are the results summarizing whether each variable or interactions between variables significantly contribute to the variance among groups.

The analysis showed that genotype, time, and FLG22 treatment affected camalexin gene expression significantly with p values lower than 0.05. The interaction between genotype and treatment was marginally significant with p value 0.062. This may be because metabolite data is nosier than the gene expression data. Another possibility is that the kinetics of metabolites might be different from transcripts since the significant accumulation of metabolite was detected at fairly late time points. We added theses caveats to the Result section reporting camalexin accumulation in response to FLG22 treatment. The interaction between treatment and time is statistically significant, which indicates the effect of FLG22 on camalexin gene induction varies based on the time of treatment. The variable Trial represents the two rounds of measurements of camalexin content in each sample. The p value for Trial is larger than 0.05, which suggests the rounds of experiments detected similar results. We further analyzed the difference of camalexin content between mock and FLG22 treated samples in different genotype by running two-way ANOVA with genotype and treatment as variables followed by Dunnett’s test.

I don't think the literature provides sufficient resolution to say that only CYP79B2 is involved in camalexin as is proposed in Figure 2. CYP79B2 KOs an induce camalexin suggesting that CYP79B3 is also involved. Similarly they CYP71A12/A13 specificity is still unresolved. I understand the focus when doing the CHIP but the genomic data should have information about these other genes and could be provided in the supplemental.

We revised Figure 2 to include CYP79B2, CYP79B3, CYP71A12, and CYP71A13 to the proposed pathway diagram. We selected CYP79B2, CYP71A13, and PAD3 in this study because these three genes are sufficient to produce camalexin in in vitro enzymatic assays (Klein et al., (2013) Angew Chem Int Ed Engl). Nonetheless, additional enzymes may play important roles in producing camalexin in vivo under particular conditions.

To respond to the reviewer’s comment, we plotted the expression of CYP79B3 and CYP91A12 and analyzed their gene expression change using one-way ANOVA in (Author response image 1). Post-hoc Tukey’s test shows that both genes showed significant induction in *pkl-1* and *Col-0*, but not in *idm1* plants, 30min after FLG22 treatment, which is consistent with the three essential genes we present in the manuscript.

**Author response image 1. sa2fig1:** 

The metabolomics in response to FLG22 was intended to show that the gene expression changes were indicative of metabolite changes. Unfortunately, the MS finds largely unknown compounds which could be all related to camalexin synthesis or other CYP79B2 dependent metabolites. As such, it doesn't really support the broader argument about all specialized metabolites. There is also the chance that the enhanced response in clf28 could be due to lower basal levels. Right now with FC levels shown it isn't clear if this is an enhanced response or enhanced repression. It would help to show absolute levels and to conduct factorial analysis testing for genotype and genotype x treatment effects. This is a similar concern for the microarray analysis as well.

We attempted to identify additional metabolic pathways that may be controlled by both H3K27me3-H3k18ac bivalent chromatin as camalexin genes in response to FLG22 signal. We used RNA-seq and untargeted metabolomics to measure gene expression and metabolites change in different genotypes treated under FLG22 treatment. Several pathways were identified using enrichment analysis with metabolic genes showing similar gene expression patterns as camalexin genes. We feel that the RNA-seq and metabolomics data provide valuable preliminary information for future studies but divert attention away from the major finding of this manuscript on camalexin biosynthesis pathway. Thus, we removed sections reporting these results in the manuscript.

Editorial SuggestionsI'm not totally sure what the main point of the second paragraph of the discussion is conveying. This paragraph starts with ideas about bivalent chromatin in development and then ends with specific details about bivalent chromatin in camalexin. I feel there is a big point in this paragraph but as written I can't quite grasp it. eLife allows more space then is used and it would help the general reader to expand this section to let the big point come through clearer. I think it is about bivalent chromatin being a trigger switch when something must go from absolute off to absolute on and do so very quickly.

We appreciate the reviewer’s comment about improving the section of Discussion on the function of bivalent chromatin. In the original submission, we discussed the observation of bivalent chromatin in the literature and its biological function discovered this study in the second paragraph of Discussion. To better highlight the knowledge gap filled in this study in understanding the function of bivalent chromatin, we rewrote this section to discuss the known knowledge on bivalent chromatin function and the main discovery of this manuscript. Here are the revised paragraphs:

“The function of bivalent chromatin has long been hypothesized to poise developmental genes in embryonic stem cells for rapid activation upon a cell differentiation signal^22^. Only few studies directly tested this hypothesis and all of them were focusing on stem cells or cancer cells^47,48,49^. For example, in embryonic stem cells, H3K27me3-H3K4me3 bivalent chromatin was associated with *HoxB4*, one of the key regulators controlling development. Inhibition of H3K4me methyltransferase, hSET1A, repressed the expression of HoxB4 and led to a block in the differentiation of blood cells^47^. In addition, H3K27me3-H3K4me3 bivalent chromatin was observed in cancer-initiating cells. Disruption of the bivalent state through inhibition of the H3K27 methyltransferase EZH2 inhibited the self-renewal of cancer cells through de-repression of a key canonical marker of normal colonocyte differentiation, named Indian Hedgehog^48^. Despite this knowledge, the role of bivalent chromatin on regulating metabolism remains unknown. In this study, we reported the colocalized of H3K27me3 and H3K18ac on camalexin genes and functionally examined the role of H3K27me3-H3K18ac bivalent chromatin in Arabidopsis. The results suggest that H3K27me3-H3K18ac bivalent chromatin maintains the proper timing of camalexin gene induction upon a pathogen signal.”

Related to the general idea of bivalent chromatin and camalexin, there are some complications around this pathway that aren't being integrated into the discussion. It doesn't change the results of this work but it does lay out a more complicated view then is being proposed. For example, the PAD mutants from Jane Glazebrook were isolated based on the accumulation of camalexin. A result of this work was the fact that the regulatory mutants (e.g. PAD4, etc) were highly specific to the stimulus being applied and often did not affect camalexin regulation in response to other stimuli. There is also work showing that the camalexin pathway has post-transcriptional regulation where the transcripts can accumulate to a high level with no metabolite accumulation. Finally, the FLG22 ability to induce camalexin is age dependent and is not seen in older soil grown plants based on the MAPK studies. Thus, the upstream and downstream regulatory processes around camalexin are significantly more complicated than the intro and discussion lays out by describing a couple of TFs. It would help to expand the discussion and introduction to convey this full complexity that is present in the established literature as it is critical to place the bivalent chromatin into the full context of camalexin regulation.

We thank the reviewer for pointing this out and helping us improve the manuscript by providing a more holistic context of camalexin regulation. We revised Introduction and Discussion to include current understanding of post-transcriptional regulation on camalexin pathway and interpret the results in a more holistic view. Revisions in the Introduction:

“Previous studies revealed complex transcriptional and translational control of camalexin biosynthesis genes^8, 9, 10, 11, 12, 13^. At the transcriptional regulation level, transcription factors from the MYB family, including MYB34, MYB51 and MYB122 promote camalexin biosynthesis in response to *P. syringae* infection^9, 12^. WRKY33 functions as an activator and directly binds to the promoters of camalexin biosynthesis genes^11^. Besides transcription factors, CALCIUM-DEPENDENT PROTEIN KINASE (CPK)5/6 and MAPK3/6 can phosphorylate WRKY33 to enhance promoter binding and transactivation^10, 13^. At the translational level, ribosome footprinting showed that genes involved in camalexin biosynthesis, CYP79B2 and CYP79B3, also increased translational efficiency under pattern triggered immunity^14, 15^. Despite the rich knowledge of these upstream regulators of the camalexin biosynthetic pathway, it remains unknown how the rapid induction upon a pathogen signal is enabled”

In Discussion, we added:

“Our results provide new evidence for how chemical defense mediated by camalexin may be regulated at the epigenetic level. However, we cannot rule out the possibility that other known mechanisms regulating camalexin genes may also affect the transcription kinetics and metabolite accumulation. For example, H3K27me3 affects gene expression by altering chromatin accessibility to transcription factors ^21^. Removing this repression mark may create a permissive environment and facilitate transcription factors, such as WRKY33, to bind to promoters of camalexin genes. At the translational level, camalexin biosynthesis genes can alter translational efficiency controlled by a highly enriched messenger RNA consensus sequence, R-motif, during pattern triggered immunity ^14, 15^. Additional studies are needed to unravel how different regulatory machineries work together to enable the rapid induction of camalexin genes upon stress signals.”

Reviewer #3 (Recommendations for the authors):The genes being studied are targeted by the PRC2 complex and this is why they have H3K27me3. Past studies have discovered that H3K18ac and H2A.Z colocalize with H3K27me3 (Luo C, 2012). Yet, there is no mention of H2A.Z throughout this study? Would this now make this chromatin state 'trivalent' chromatin? This issue specifically highlights the incorrect premise of this study, that there is a "bivalent chromatin" at these metabolic genes. They, like many other H3K27me3 marked genes, are targets of PRC2 and silenced until proper environmental/development cues are perceived. This fits with the highly specialized need for these metabolic genes in unique environments.

Luo et al., (2012) The Plant Journal did not study H2A.Z and there is no information about H2A.Z colocalized with H3K27me3 in that publication. In addition, we respectively disagree with the reviewer that H2A.Z is known to colocalize with H3K27me3. In mammalian embryonic stem cells and neural progenitors, H2A.Z is deposited at promoters marked by H3K4me3-H3K27me3 bivalent chromatin and at active promoters solely marked by H3K4me3, but not the chromatin solely associated with only H3K27me3 (Ku et al., (2012) Genome Biology). In mouse embryonic stem cells, H3K27me3 enrichment correlates strongly with H2A.Z based on ChIP-seq profiles, which indicates the co-localization of H2A.Z and H3K27me3 (Wang et al. (2018) BMC Biology). In Arabidopsis, H2A.Z and H3K27me3 have overlapping targets, such as genes associated with thalianol biosynthesis, but the co-localized has not been experimentally examined (Sequeira-Mendes et al., (2014) Plant Cell, Carter et al., (2018) Plant Cell, Nützmann et al., (2015) New Phytologist). These results suggest that H3K27me3 and H1A.Z may have different relationships in different organisms and their colocalization in Arabidopsis has not been tested.

There is no evidence presented that H3K18ac is an 'activating" mark. This should not be confused with other histone acetylation marks. Based on what was previously described by Luo C et al., 2012, it is a repressive mark given it's colocalization with H3K27me3.

We disagree with the reviewer that H3K18ac was reported as a repression mark in Luo et al., (2012) The Plant Journal. We cited the sentence describing H3K18ac-H3K27me3 colocalization in this publication:

“Intriguingly, a strong correlation was detected between H3K18Ac and H3K27me3 in the Arabidopsis genome (Pearson r = 0.44, Figure 1b and Figure S4), although histone acetylation is not expected to co‐localize with a repressive mark such as H3K27me3.”

Indeed, H3K18ac was reported as an activation mark in Luo et al., (2012) The Plant Journal, which is cited here “Our analysis also showed that multiple active histone marks such as H3K4me3, H3K9Ac and H3K18Ac were significantly more depleted in gene bodies for genes in States 2, 3 and 4 compared to those in State 1 (Figure 5f–h and Figure S9b–d)”.

There are additional reports on H3K18ac functioning as an activation mark, such as Liu et al., (2013) The Plant Cell, “Among the seven known Lys residues available for acetylation in the H3 N-terminal tail, acetylated Lys-9, Lys^-1^4, Lys^-1^8, and Lys-23 are responsible for promoting transcription”. In addition, studies in other systems, such human cell lines and *Plasmodium falciparum* also revealed that H3K18ac promotes activation of target genes, such as Luo et al., (2019) Stem Cell Reports, Tang et al., (2020) Epigenetics and Chromatin, Wang et al., (2008) Nature Genetics, Roy et al., (2017) Genome Research.

How do you distinguish direct versus indirect effects when using pkl/clf/idm1 mutants? These genes are required for general gene regulation. For example, without clf, no H3K27me3 occurs leading to up-regulation of all PRC2 targeted genes?

The reviewer brought up a good point that it’s hard to distinguish the direct and indirect effects in the mutants with defective epigenetic modifications, which is also a challenge in the field. We used mutant lines in two independent genes for each mark to test the function of H3K27me3 (*pkl*, *clf*) and H3K18ac (*idm1, idm2*), which revealed similar results (Figure 3 A to D). In addition, the up-regulation of camalexin genes is associated with the alteration of these two marks (Figure 3 E to F). These results support that epigenetic modifications do play a role to regulate the expression of camalexin genes.

I have major concerns that IDM1 is the HAT that acetylates H3K18 in gene bodies. IDM1 specifically binds to methylated DNA, which does not colocalize with H3K27me3 in Arabidopsis (see also Figure 1a). This would also explain the seemingly weak signals from all experiments using idm1. ChIP-seq data for IDM1 exists (https://www.ncbi.nlm.nih.gov/pmc/articles/PMC3575687/) and should be used to show it colocalizes specifically to the genes that have H3K27me3 versus other types of genes.

We double checked the publication highlighted by the reviewer regarding ChIP-seq data in idm1 and this study only performed ChIP-PCR on target genes instead of ChIP-seq using idm1 mutant. So unfortunately, no ChIP-seq data from the paper could be obtained. In one of the follow-up studies from the same group that discovered IDM1 (Nie et al., (2019) PNAS), H3K18ac was profiled genome-wide using *imd1* mutant in the background overexpressing SUC2. There, this dataset is also not appropriate for our study.

IDM1 is a histone acetyltransferase, which binds to DNA methylated region to initiate DNA demethylation process by recruiting SWR1 and ROS1 protein complexes (Qian et al., (2014) Molecular Cell, Nie et al., (2019) PNAS). We checked the DNA methylation pattern of camalexin genes using the methylome profiles generated by 1001 genomes project (http://neomorph.salk.edu/1001.aj.php) and the three genes we tested in this study have DNA methylation in the gene body. This suggests that camalexin genes can be targets of IDM1.

In addition, the enzyme complex required to establish H3K18ac is not well characterized, which limits the genetic resources we can use. We revised the text to reflect the challenge and limitations to dissect the function of H3K18ac on the induction of camalexin genes: “The protein complex involved in establishing H3K18ac has not been well characterized in Arabidopsis, which limits the mutant lines that can be utilized to study the function of this epigenetic modification. The functionally characterized mutant lines associated with establishing H3K18ac are increased DNA methylation 1 and 2 (idm1 and idm2)^33, 34^. IDM1 is the histone acetyltransferase and IDM2 is a heat shock protein that functions in the same protein complex as IDM1.”

The term bivalent chromatin should be more carefully used in this study. Bivalent chromatin specifically refers to H3K4me3/H3K27me3. It should not be used to described H3K27me3/H3K18ac. There are numerous examples where bivalent chromatin is referring to the classic definition (H3K4me3/H3K27me3) to set up rationale for proposed experiments on H3K27me3/H3K18ac. This leads to incorrect hypotheses and is confusing to the reader.

We disagree with the reviewer that bivalent chromatin specifically refers to H3K4me3 and H3K27me3 and we cited several references that support our definition used in the manuscript: “the bivalent domains (BDs) – that are distinguished for having two histone marks that are associated with both positive and negative transcriptional events”. Refs: Blanco et al., (2019) Trends in Genetics; Sneppen et al., (2019) Nature Communications; Zhang et al., (2020) Molecular Plant; Velde et al., (2021) Communications Biology, Jacquet et al. (2016) Molecular Cell. To avoid any confusion, we revised the title and the manuscript to clarify that we are reporting a novel type of bivalent chromatin instead of the well-studied variant formed by H3K4me3 and H3K27me3.